# Decompensated MASH-Cirrhosis Model by Acute and Toxic Effects of Phenobarbital

**DOI:** 10.3390/cells13201707

**Published:** 2024-10-16

**Authors:** Nico Kraus, Frank Erhard Uschner, Magnus Moeslein, Robert Schierwagen, Wenyi Gu, Maximilian Joseph Brol, Eike Fürst, Inga Grünewald, Sophie Lotersztajn, Pierre-Emmanuel Rautou, Marta Duran-Güell, Roger Flores Costa, Joan Clària, Jonel Trebicka, Sabine Klein

**Affiliations:** 1Department of Internal Medicine I, Hospital of the Goethe University, 60596 Frankfurt, Germany; n.kraus@med.uni-frankfurt.de (N.K.); frankerhard.uschner@ukmuenster.de (F.E.U.); magnus_moeslein@hotmail.com (M.M.); robert.schierwagen@ukmuenster.de (R.S.); wenyi.gu@ukmuenster.de (W.G.); maximilian.brol@ukmuenster.de (M.J.B.); sabine.klein@ukmuenster.de (S.K.); 2Department of Internal Medicine B, University Clinic Münster, 48149 Münster, Germany; eike.fuerst@ukmuenster.de; 3Gerhard-Domagk-Institute of Pathology, University Clinic Muenster, 48149 Münster, Germany; inga.gruenewald@ukmuenster.de; 4Centre de Recherche sur L’inflammation, Université Paris-Cité, Inserm, UMR 1149, 45018 Paris, France; sophie.lotersztajn@inserm.fr (S.L.); perautou@yahoo.fr (P.-E.R.); 5AP-HP, Hôpital Beaujon, Service d’Hépatologie, DMU DIGEST, Centre de Référence des Maladies Vasculaires du Foie, FILFOIE, ERN RARE-LIVER, 03200 Clichy, France; 6Department of Biochemistry/Molecular Genetics, Hospital Clinic de Barcelona, IDIBAPS and CIBERehd, 08028 Barcelona, Spain; mduran@cytesbiotech.com (M.D.-G.); roger.flores@hipra.com (R.F.C.); jclaria@clinic.cat (J.C.); 7Department of Biomedical Sciences, University of Barcelona, 08036 Barcelona, Spain; 8European Foundation for the Study of Chronic Liver Failure, 08021 Barcelona, Spain

**Keywords:** metabolic dysfunction-associated steatohepatitis (MASH), carbon tetrachloride (CCl_4_), high-fat and high-cholesterol Western diet (WD), phenobarbital (PB), long-term (LT), short-term (ST)

## Abstract

Metabolic dysfunction-associated Steatohepatitis (MASH), is a prominent cause for liver cirrhosis. MASH-cirrhosis is responsible for liver complications and there is no specific treatment. To develop new therapeutic approaches, animal models are needed. The aim of this study was to develop a fast animal model of MASH-cirrhosis in rats reflecting the human disease. Carbon tetrachloride (CCl_4_) injections in combination with a high-fat Western diet (WD) were used to induce MASH-cirrhosis. To accelerate liver injury, animals received phenobarbital (PB) in their drinking water using two different regimens. Rats developed advanced MASH-cirrhosis characterized by portal hypertension, blood biochemistry, hepatic ballooning, steatosis, inflammation and fibrosis. Importantly, rats receiving low-dose PB for the long term (LT) showed ascites after 6 weeks, whereas rats with high-dose short-term (ST) PB developed ascites after 8 weeks. ST- and LT-treated rats showed increased portal pressure (PP) and decreased mean arterial pressure (MAP). Of note, hepatocyte ballooning was only observed in the LT group. The LT administration of low-dose PB with CCl_4_ intoxication and WD represents a fast and reproducible rat model mimicking decompensated MASH-cirrhosis in humans. Thus, CCl_4_ + WD with LT low-dose phenobarbital treatment might be the preferred rat animal model for drug development in MASH-cirrhosis.

## 1. Introduction

Steatotic liver disease is an emerging health problem world-wide caused by metabolic, toxic or genetic injury or combinations of these factors [1,2]. According to the main insult, steatotic liver disease may develop from either alcohol- or non-alcohol-related liver injury, and hospital admissions of patients with non-alcohol-related cirrhosis has increased four times in the last decade [3].

Metabolic dysfunction-associated steatotic liver disease (MASLD), previously known as non-alcohol-related fatty liver disease (NAFLD), is considered as a new epidemic of chronic liver disease [4]. The prevalence is estimated to be up to 25% in the general population and even higher in patients with obesity and diabetes [5,6]. One third of MASLD patients progress and develop metabolic dysfunction-associated steatohepatitis (MASH), a form of steatotic liver disease with progressive hepatic fibrogenesis. MASH can be the precursor of liver cirrhosis, hepatocellular carcinoma (HCC) and liver-related mortality and is, therefore, an increasing indication for liver transplantation in Western countries [7,8]. To address the increasing MASLD incidence, safe and effective therapeutics for patients with advanced liver fibrosis are needed.

Existing animal models do not fully recapitulate MASH-cirrhosis in humans and limit the development of new therapeutic strategies [9,10].

Most experimental models are developed to study early stages of MASLD, since they mimic the characteristic features of MASLD including steatosis, hepatocyte ballooning and inflammation, but they usually lack hepatic fibrosis [1,2,7]. At the time of diagnosis, portal hypertension is already manifested in 25% of MASH patients [11] and patients with advanced MASH have a higher prevalence of portal hypertension than other etiologies [12]. Therefore, studies of MASH-cirrhosis should be performed in animal models showing features of significant portal hypertension. Unfortunately, all existing rat models of advanced MASH need long-lasting protocols to establish severe portal hypertension and its complication [9,13]. Thus, faster and better defined animal models are needed to hasten drug development in MASH-cirrhosis.

This study aims to accelerate an existing rodent MASH-cirrhosis model to mimic advanced human MASH.

## 2. Materials and Methods

### 2.1. Animals

Six-week-old male *Sprague Dawley (SD)* rats (Charles River Laboratories Research Model and Services Germany, Sulzfeld, Germany) were housed in a 12:12 h light–dark cycle, with a controlled temperature (22 °C ± 2 °C) and 50% humidity. Healthy and CCl_4_-intoxicated animals had ad libitum access to standard rat chow (Ssniff, Soest, Germany), while control (CCl_4_ + WD), ST- and LT-treated animals had access to a high-fat and high-cholesterol Western diet (WD) (containing 20.9% crude fat and 1.25% cholesterol). All animals had free access to drinking water.

A total of 51 SD rats were used, separated in healthy (n = 5), CCl_4_ (n = 5), control (CCl_4_ + WD) (n = 6), ST treatment week 1 (n = 10), ST treatment week 2 (n = 10), ST treatment week 3 (n = 10) and LT treatment (n = 5) groups.

All animal experiments were approved by the German animal welfare Federation of the regional council Hesse (no. FK/2003) and of the regional council Northrhine Westfalia (81-02.04.2022.A334) conducted in accordance with the German animal protection and welfare law and the guidelines of the animal care facility at the university clinic in Frankfurt and at the university clinic in Muenster.

### 2.2. CCl_4_ Intoxication

To induce toxic liver cirrhosis, rats were injected i.p. with CCl_4_ diluted in Corn Oil (1:1) twice per week (Sigma Aldrich, Taufkirchen, Germany). For the first two weeks, the rats received 2 µL/g and, thereafter, 1 µL/g of CCl_4_, respectively_._ Rats were treated for 20 weeks or until ascites was present, as a sign of portal hypertension.

### 2.3. Phenobarbital (PB) Administration

Simultaneously with CCl_4_ injections, rats with ST treatment received 0.3 g/L PB (Luminal^®^, DESITIN, Hamburg, Germany) in the drinking water for 3 consecutive days in the first week, the second week or in the third week and, thereafter, normal drinking water until ascites developed. Rats with LT treatment received permanently 0.06 g/L PB in their drinking water until ascites developed. Treatment with PB or CCL_4_ was terminated in all animals for 7 days prior to the organ harvest and functional tests.

### 2.4. Biochemical Measurements

To assess the liver function, Aspartate aminotransferase (AST), Alanine aminotransferase (ALT), total Bilirubin (tBIL), Albumin (ALB) and Gamma-Glutamyl Transferase (γGT) were analyzed in Li-Heparin plasma (S-Monovette LH, Sarstedt, Nümbrecht, Germany) using the Fuji Drychem NX 500i (Scilvet, Weinheim, Germany).

The Plasma Triglycerides concentration was measured using the Triglycerides liquicolor mono kit (Human, Wiesbaden, Germany).

### 2.5. Liver Histology

Organ harvest was performed when ascites was present and after an additional period of 7 days of treatment discontinuation. Liver samples were snap-frozen and stored at −80 °C or fixed in formaldehyde. After fixation they were embedded in paraffin or fixed in Tissue Tek OCT (Sakura Finetek Germany, Umkirch, Germany). For histology, liver samples were fixed in 4% formalin for 24 h, embedded in paraffin, sectioned (2–3 µm) and slides were stained with hematoxylin and eosin (H&E) (Thermo Scientific, Runcorn, UK). To evaluate fibrosis, liver sections were stained with Sirius red (SR) (0.1% saturated picric acid). α-smooth muscle actin (αSMA) was stained immunohistochemically (IHC) (A2547, Sigma-Aldrich, Taufkirchen, Germany) to visualize the activation of HSC (Chroma, Münster, Germany). Cryosections (5–10 μm) of liver samples were fixed and stained with Oil red O, as described previously [14].

Stainings were acquired with a Nikon Digital Sight DS-Vi1 (Chiyoda, Tokyo, Japan) microscope and quantified with the open-source ImageJ software (V.1.51j8; National Institutes of Health, Bethesda, MD, USA).

### 2.6. Fibrosis Scoring

Liver fibrosis was assessed by a pathologist, blinded to the different treatment regimens, using an established fibrosis score according to Kleiner et al. [15]. The following stages of fibrosis were differentiated according to the literature: F0 = no fibrosis; F1 = perisinusoidal or periportal fibrosis; F2 = perisinusoidal and periportal fibrosis; F3 = bridging fibrosis; F4 = cirrhosis. The histological scoring was performed in the Sirius red-stained livers of all rats.

### 2.7. Western Blotting

Protein expression was analyzed through Western blot analyses as described previously [16]. Briefly, snap-frozen livers were homogenized and diluted before the protein concentration of the homogenates was determined using Bradford (Bio-Rad, Munich, Germany). Samples (25 µg of protein/lane) were loaded to SDS-PAGE (10% gels). Thereafter, the proteins were blotted on nitrocellulose membranes as the endogenous control served Glyceraldehyde-3phosphate dehydrogenase (GAPDH) and blots were developed using enhanced chemiluminescence. The used antibodies are listed in the Appendix A.

### 2.8. Hydroxyproline Content Measurement

To assess fibrosis, the hepatic hydroxyproline (HP) content was measured. For this, 250–300 mg liver tissues from two different lobes (representing >10% of the liver) were measured biochemically. The total HP (μg/g liver) was determined based on individual liver weights and the corresponding relative HP content as described previously [17].

### 2.9. In Vivo Portal and Systemic Pressure Measurements

Pressure measurements were performed in anesthetized rats (ketamine 100 mg/kg/xylacine 10 mg/kg) as previously described [18]. Briefly, a midline laparotomy was performed and a PE-50 catheter was inserted into a small ileocecal vein and advanced to the portal vein to measure the portal pressure (PP; mmHg). The mean arterial pressure (MAP; mmHg) was measured via a PE-50 catheter in the left femoral artery. Both catheters, in the femoral artery and the portal vein, were connected to pressure transducers (ADInstruments Ltd., Oxford, UK). After the rats were hemodynamically stabilized, the PP and MAP were measured.

### 2.10. Brain Injury Assessment

To evaluate hepatic encephalopathy due to liver cirrhosis, the brain water content was measured. For this, a small piece (5 mm^3^) of the frontal lobe was weighted (brain wet weight) and dried at 95 °C for 24 h. The brain water content was calculated as the percentage of the brain dry weight and brain wet weight.

Additionally, the Revised Neurobehavioral Severity Score for rodents was performed after 7 days of treatment discontinuation. This scoring system contains ten tests of specific, sensitive and standardized observations to analyze the balance, motor coordination and sensorimotor reflection in rats, as described previously [19].

### 2.11. Quantitative Real-Time PCR

To measure gene expressions, a quantitative real-time PCR was performed as described previously [20]. For this, the total RNA was isolated using a customary Trizol (TRIzol Reagent, Ambion, Carlsbad, CA, USA)-based protocol. The cDNA was synthesized with the ImProm-II Reverse Transcription System (Promega, Madison, WI, USA). The qPCR was performed with a StepOnePlus Real-Time PCR System (Applied Biosystems, Foster City, CA, USA) using TaqMan gene expression assays (Thermo Fisher Scientific, Waltham, MA, USA) and the manufacturer’s protocol. All experiments were executed in duplicates. The relative gene expression was quantified by the 2^−ΔΔCt^ method. The results were normalized to the endogenous control *18S rRNA* expression. The full list of TaqMan gene expression assays is provided in Appendix A.

### 2.12. Statistical Analysis

All statistical analyses were conducted using Prism V.10.0 (GraphPad, San Diego, CA, USA). The results were shown as means ± SEM, and comparisons between two groups were performed by a nonparametric Mann–Whitney U test. *p* values < 0.05 were considered significant.

## 3. Results

### 3.1. Development of Portal Hypertension and Disease-Dependent Mortality

To accelerate the progression of the disease, advanced MASH-cirrhosis rats received CCl_4_ with WD in combination with high-dose PB (0.3 g/L) for 3 days (ST) in their drinking water or low-dose PB (0.06 g/L) continuously (LT) in their drinking water until ascites developed (Figure 1A). These PB-treated rats were compared to healthy rats, CCl_4_-intoxicated rats (CCl_4_) and rats with CCl_4_ and WD, but without PB treatment (control).

The ST treatment showed high acute mortality within 3 days of PB administration, independent of the chosen administration week (week 1, week 2 or week 3), compared to healthy, CCl_4_-intoxicated and CCl_4_ + WD control rats (Figure 1B). The rats of the LT-treated group showed less acute mortality than the ST-treated group (Figure 1C).

The LT group developed ascites already after 4 to 6 weeks and the ST-treated rats after 8 weeks, while the CCl_4_ as well as CCl_4_ + WD rats developed ascites after 17 weeks and 12 weeks, respectively (Figure 1D).

In order to assess the severity of ascites development, ascites was collected and the amount was measured before sacrifice. The ascites volume was classified as moderate (++), mild (+) and no (−) ascites. All rats treated with PB showed a higher amount of ascites in comparison to the CCl_4_ and the CCl_4_ + WD groups without PB (control), who developed only mild ascites (Figure 1E). Importantly, this effect was observed after a shorter duration of cirrhosis induction in LT and ST compared to CCl_4_ and CCl_4_ + WD. Nevertheless, we could not observe differences between the ST or LT treatment.

To assess portal hypertension invasively, PP was measured in vivo. In all cirrhosis models, rats developed significantly increased PP compared to healthy controls. Even though there was no significant difference between the cirrhosis groups, LT and ST animals developed portal hypertension in a shorter time period than CCl_4_ or CCl_4_ + WD animals (Figure 1F). Simultaneously, the MAP was reduced, due to decreased systemic circulation and splanchnic vasodilatation. However, MAP was not different in CCl_4_ + WD, ST and LT rats compared to CCl_4_ alone (Figure 1G).

Characterization of chronic liver injury. To investigate the structural and cellular hepatic changes, Hematoxylin and Eosin (H&E) staining was performed. Only after LT treatment was relevant hepatocyte ballooning observed, classified by an enlarged hepatocyte volume and visible lobular inflammation, as shown by the black arrows in the 40× magnification (Figure 2A).

Fibrosis progression was evaluated in Sirius red-stained liver sections using established fibrosis stages F0–F4. The majority of healthy controls were scored as F0 (80%), while 50% of the CCl_4_-injected rats showed features of F4 fibrosis. Of note, all analyzed sections of animals treated with ST and LT phenobarbital were accounted for as F4 fibrosis, while only 65% of the control rats had a similar severity of fibrosis, as analyzed by Sirius red staining (Figure 2B).

Hepatic collagen deposition was investigated by Sirius red (SR) staining in all rats. All treated animals showed the characteristic bridging fibrosis compared to healthy controls. Hepatic SR staining revealed increased collagen deposition after ST or LT treatment, as shown in the corresponding image and quantification (Figure 2C,D). This finding was confirmed by a significantly elevated HP content in liver samples of ST- and LT-treated rats compared to CCl_4_ alone (Figure 2E). However, there was no significant change in the hepatic HP content between CCl_4_ + WD rats without PB treatment and the ST- and LT-treated groups. The hepatic gene expression of *Col1a1* was increased in all CCl_4_-treated groups compared to the healthy controls. LT treatment induced significantly higher *Col1a1* expression levels compared to CCl_4_ alone (Figure 2F). The hepatic protein levels of Col1a1 were also significantly increased in the ST and LT groups compared to the healthy controls (Figure 2G).

To examine hepatic stellate cells (HSC)’s activation, αSMA staining and gene and protein levels were analyzed. After ST and LT treatment, αSMA deposition was increased compared to the healthy and CCl_4_-intoxicated animals (Figure 2H–K). This was confirmed by decreased *Gfap* mRNA expression, a marker for quiescent HSCs, in ST and LT livers (Figure 2L). Furthermore, the lowest *Gfap* gene expression was observed in the ST-treated animals, which further supports the strong and acute toxic effect of high-dose ST treatment compared to LT and CCl_4_ + WD treatment (Figure 2L). Nevertheless, the αSMA protein expression levels were not changed after ST or LT treatment compared to the control rats (CCl_4_ + WD without Phenobarbital) (Figure 2K). Overall, these results demonstrate that ST and LT treatment significantly increases HSC activation and hepatic collagen deposition.

### 3.2. Hepatic Steatosis and Inflammation after ST and LT Treatment

The presence of hepatic steatosis was investigated by hepatic Oil red O staining and the measurement of the triglyceride content in the plasmas. Oil red O-stained areas were significantly increased in all cirrhotic rats, but most pronounced in ST and LT liver samples (Figure 3A). Moreover, plasma triglyceride levels were significantly increased after ST and LT treatment compared to healthy, CCl_4_ and CCl_4_ + WD control rats (Figure 3B).

Next, the hepatic expression levels of the key protein involved in lipogenesis, the sterol regulatory element binding protein-1c (SREBP-1c), was analyzed. The protein expression of SREBP-1c was significantly changed in CCl_4_ rats, CCl_4_ + WD control rats and even more after ST and LT treatments with PB (Figure 3C). To investigate hepatocyte injury, ALT and AST were measured in the plasmas of all rats (Figure 3D,E). ALT levels were significantly increased after CCl_4_ + WD, ST and LT treatment compared to healthy and CCl_4_-intoxicated rats (Figure 3D). Moreover, AST levels were significantly increased after CCl_4_ + WD and ST treatment, but remained almost within normal limits after LT treatment. In line, the De Ritis ratio was reduced in LT animals, suggesting a more advanced stage of liver cirrhosis (Figure 3F) [21]. However, γGT levels were only increased after LT treatment compared to healthy and cirrhotic animal groups (Figure 3G).

To assess hepatic inflammation, the gene expression of the pro-inflammatory markers *IL6* and *CCL2* were measured (Figure 3H,J). Hepatic inflammation was significantly increased after ST and LT treatment, shown by increased gene expression levels of *CCL2* compared to healthy and CCl_4_-intoxicated rats (Figure 3J). Nevertheless, *IL6* was only slightly elevated in the different cirrhosis models. Infiltrating macrophages due to liver inflammation were detected by the EMR1 marker. *EMR1* gene expression levels were significantly increased in all cirrhosis models (Figure 3I). However, ST or LT treatment with PB showed similar results as CCl_4_ and CCl_4_ + WD animals.

PB-induced liver and brain dysfunction in experimental cirrhosis. ST- and LT-treated rats showed a tendency towards increased TBIL levels (Figure 4A) and decreased Albumin levels in serum, suggesting evident liver dysfunction in both models (Figure 4B). Furthermore, only LT rats displayed signs of coagulation failure, as shown by increased INR levels (Figure 1C).

The extent of brain impairment after ST and LT treatment was investigated by neurological behavior tests and by the measurement of the brain water content. Neurological behavior tests exhibited abnormalities after ST and LT treatment. LT treatment showed significantly increased relative units in the neurological behavior test compared to all other models (Figure 4D). This finding was confirmed by an increased brain water content after CCl_4_ + WD control, ST and LT treatments (Figure 4E), which represents a cerebral edema as a common feature associated with hepatic encephalopathy.

## 4. Discussion

In this study, we report two advanced MASH-cirrhosis models in rats, based on CCl_4_ intoxication combined with WD and cytochrome P450 induction by phenobarbital. Both models show the fast progression of liver disease with histological and biochemical features of advanced MASH, combined with portal hypertension, coagulation impairment and brain dysfunction recapitulating decompensated human MASH-cirrhosis.

MASH research is challenged by the lack of translational animal models that mimic its etiology and pathogenesis as well as reproduce the histological features of human MASH [9]. Furthermore, only a few animal models develop full-blown cirrhosis and clinically significant portal hypertension [22]. Although fibrosis is not strictly part of the histological definition of MASH, it is one of the best predictors of liver-related mortality in patients [22,23]. With disease progression, central veins and portal areas are linked by fibrous bridges, hallmarking the stage of advanced fibrosis (bridging fibrosis) and resulting in cirrhosis, which is characterized by fibrotic septa that link portal and central areas and form nodules with isolated hepatic parenchyma [24,25].

Our animal models reproduced bridging fibrosis in a shorter period of time compared to known and established animal models [9,26].

Furthermore, increased extracellular matrix deposition, as well as hepatic stellate cell activation, that is known to be crucially involved in fibrogenesis, prove the advanced stage of MASH-cirrhosis in our models [27]. Steatosis and inflammation are key histological hallmarks for human MASH and are associated with metabolic alterations, such as hypertriglyceridemia and dyslipidemia [28]. Oil red O staining clearly indicates a gradual increase in fat accumulation in our models, especially in ST- and LT-treated animals, which is in line with human MASH. Nevertheless, plasma triglyceride levels, as a surrogate of metabolic dysfunction, do not completely complement the histological data. Thus, the highest values were measured in the ST model.

Thus, our animal models of ST and LT combined with CCl_4_ and WD show histological features such as lobular inflammation and hepatocyte ballooning, an increased hepatic expression of proinflammatory cytokines and metabolic criteria of human MASH-cirrhosis [29]. Importantly, blood chemistry data of LT and ST rats demonstrate chronic liver damage that is comparable to CCl_4_ and WD alone. Interestingly, LT-treated animals had lower levels of AST and especially of ALT, which resulted in a lower De Ritis ratio compared to ST and CCl_4_ + WD animals. This, in turn, might be explained by a more advanced stage of liver cirrhosis [21].

Portal hypertension is a major complication of advanced liver disease. However, most animal models of MASH-cirrhosis do not develop this important clinical complication. The acceleration of MASH-cirrhosis by PB leads to severe clinically significant portal hypertension, that is reflected by earlier decompensation and higher amounts of ascites at the time of decompensation compared to CCl_4_ and CCl_4_ + WD alone. This underlines the strong hepatotoxic effect of PB treatment, which is known to induce the isozyme activities of cytochrome P-450 [30]. Furthermore, animals had signs of portal hypertension-associated hyperdynamic circulation, since the mean arterial pressure was reduced in all groups. A hallmark of decompensated cirrhosis is the development of organ dysfunction (e.g., hepatorenal syndrome or hepatic encephalopathy), that might result in acute-on-chronic liver failure, with high short-term mortality. Severe liver dysfunction was present in all cirrhotic animals, as shown by increased bilirubin levels, but only LT treatment exhibited signs of coagulation failure with an increased INR.

The combination of CCl_4_ and WD with LT and ST treatment induces extrahepatic complications, as shown by the neurological behavior test. We could observe major neurological impairment, especially in LT-treated animals, as shown by the neurological behavior test. Nevertheless, a minor influence of PB on neurological behavior, even after treatment cessation for 7 days, cannot be excluded. Therefore, we measured the brain water content and could observe that the control rats (CCl_4_ and WD) and the ST and LT animals had a significant increase in their brain water content, suggesting a brain edema and, subsequently, hepatic encephalopathy as the cause of the impaired neurological behavior [31,32].

This study also shows that a dose of 0.3 g/L PB combined with WD and CCl_4_ intoxication leads to higher mortality rates compared to CCl_4_ and WD alone. This toxic effect is dose-dependent and a dose reduction ameliorated the survival rate in the LT group. The combination of CCl_4_, PB and WD directly and acutely affects the liver function, which can be explained by the prolonged rise in the amount of cytochrome P 450 (CYP2E2) expressed in the liver, which metabolizes CCl_4_ into the toxic trichloromethyl (CCl_3_^−^) radical [33].

In our hands, the ST model was more severe and an acceleration of the PB dose and duration resulted in a hepatotoxic phenotype. Thus, the ST model might be suitable to investigate mechanisms of drug-induced liver injury in already-established chronic liver disease, but can be chronified and used to establish MASH-cirrhosis within 8 weeks. In contrast, the LT model showed all criteria of MASH-cirrhosis, with the fast development of severe portal hypertension within only 6 weeks, but this had less acute toxic effects.

A limitation of this study is a rather high mortality rate in the ST group, even though the addition of PB significantly reduced the time to cirrhosis and ascites development. Furthermore, our MASH models incorporate two different etiological factors, drug-induced and metabolic liver damage, leading to steatotic liver cirrhosis, but no MASH by definition. Tsuchida et al. already published a murine NASH, fibrosis and HCC model by CCl4, a Western diet and a high sugar solution in the drinking water [26]. They observed that after 12 and 24 weeks, the mice developed steatosis and inflammation. This model might reflect human disease more precisely compared to our model, but has the disadvantage of long treatment durations that we tried to overcome by using phenobarbital instead of a sugar solution. Nevertheless, also in human MASH, other co-factors are frequently involved and accelerate liver damage. Thus, the current nomenclature of MASLD and MASH was already adapted, recognizing co-factors of steatotic liver diseases, such as MetALD. Therefore, drug-induced liver damage by phenobarbital, as a co-factor of metabolic disease, might be in line with human MASH.

## 5. Conclusions

In summary, we have generated novel rat models of advanced MASH-cirrhosis by using CCl_4_ and WD combined with PB at a low or high dose, with key histological and pathophysiological features of decompensated human MASH-cirrhosis within a shorter period of time compared to established models of MASH. Our models are suitable as an experimental tool for the development of new pharmacological interventions for MASH-cirrhosis.

## Figures and Tables

**Figure 1 cells-13-01707-f001:**
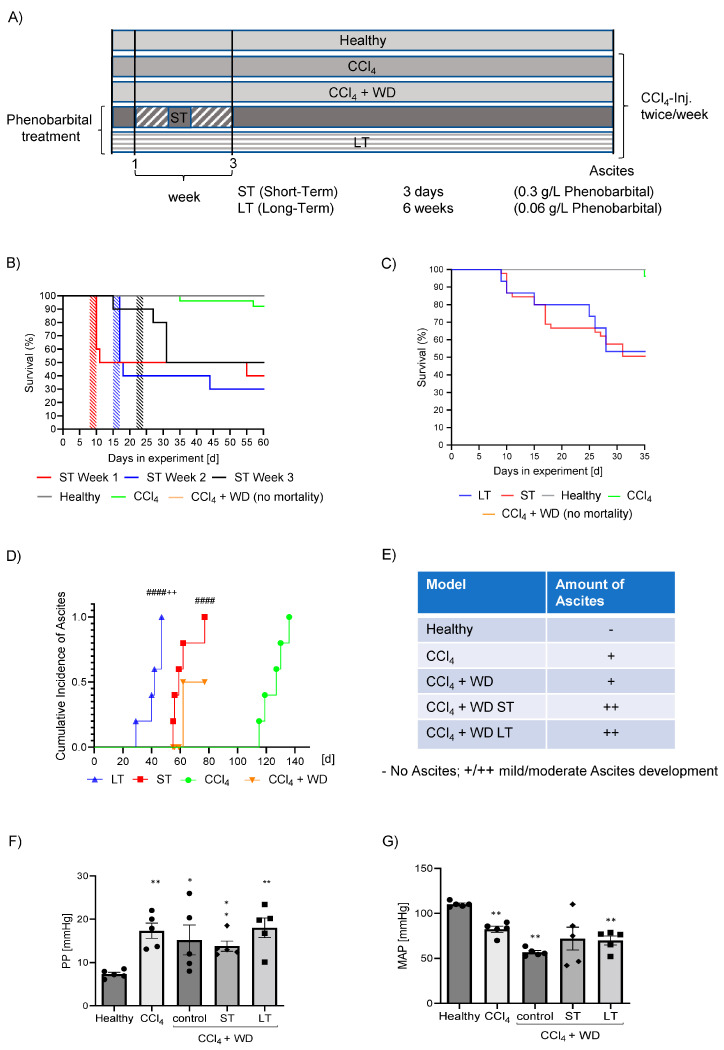
Development of portal hypertension and disease-dependent mortality. Schematic design of in vivo experiments in SD rats showing 5 groups of rats which were used. Besides healthy control rats, carbon tetrachloride (CCl_4_)-intoxicated rats and MASH rats (CCl4 + WD) without phenobarbital in their drinking water were used. Two groups of phenobarbital-treated rats were analyzed. One group with short-term (ST) and high-dose phenobarbital (0.3 g/L) treatment for 3 days and one group with long-term and low dose of phenobarbital (0.06 g/L) were used. CCl_4_ was intraperitoneally injected (i.p.) twice per week until ascites was present (**A**). Kapplan–Meier curve of all rat groups showing the survival of ST-treated rats during the first week, the second week and the third week in comparison to healthy, CCl_4_- and CCl_4_ + WD-treated rats (**B**). Survival curve of LT, ST (data pooled), healthy, CCl_4_ and CCl_4_ + WD rats (**C**). Cumulative incidence of ascites development in LT, ST, CCl_4_- and CCl_4_ + WD-treated animals in days (**D**). CCl_4_ + WD rats were not treated until the highest incidence of ascites to obtain a better comparison to phenobarbital-treated animals. Table of the models and the amount of ascites produced in the rat groups (**E**). Portal pressure (PP) of all rat groups (**F**). Mean arterial pressure (MAP) of all rat groups (**G**). Results are expressed as means + SEM; n = 5 per group. (*/** *p* < 0.05/*p* <0.001 vs. Healthy; #### *p* < 0.00001 vs. CCl_4_; ++ *p* < 0.005 vs. ST).

**Figure 2 cells-13-01707-f002:**
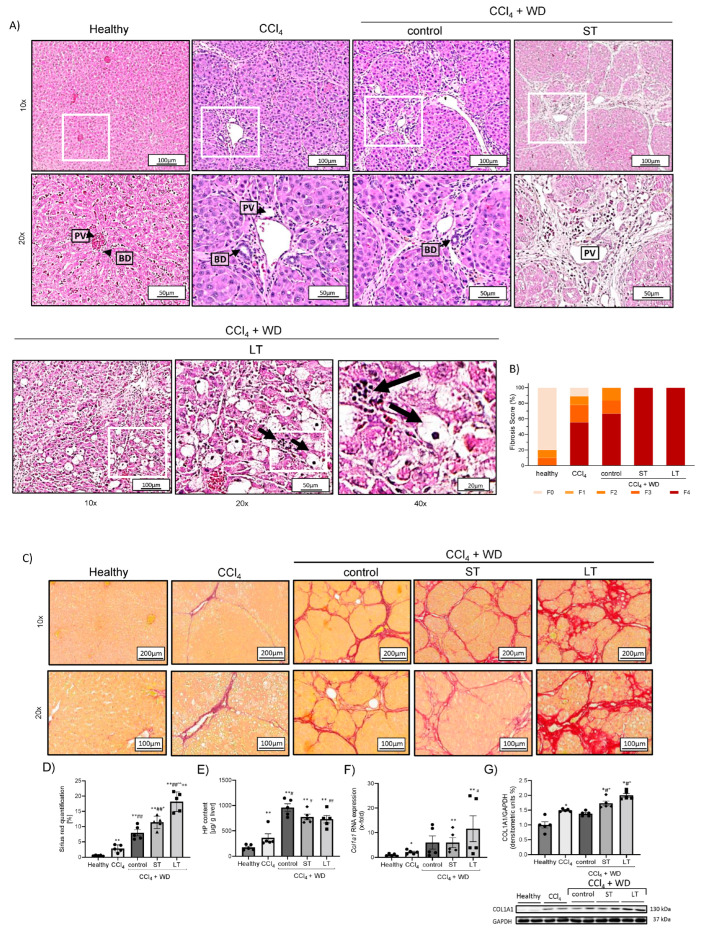
Characterization of chronic liver injury. Hematoxylin and Eosin (H&E) stainings in livers of healthy, CCl_4_, control (CCl_4_ + WD), ST- and LT-treated rats (**A**). Illustrations of the Glisson’s triad to show hepatocyte ballooning and inflammation in LT-treated livers. Inflammation and hepatocyte ballooning is indicated by arrows in 40× magnification (PV = portal vein; BD = bile duct). Hepatic fibrosis score of all rat models from grade F0–F4. Fibrosis is scored into the following groups: F0 = no fibrosis; F1 = perisinusoidal or periportal fibrosis; F2 = perisinusoidal and periportal fibrosis; F3 = bridging fibrosis; F4 = cirrhosis (**B**). Hepatic Sirius red staining and its quantification of healthy, CCl_4_, control (CCl_4_ + WD), ST- and LT-treated rats (**C**,**D**). Amount of hepatic hydroxyproline (HP) in healthy, CCl_4_, control (CCl_4_ + WD), ST- and LT-treated rats (**E**). Gene expression level of *Col1a1* in livers of healthy, CCl_4_, control (CCl_4_ + WD), ST- and LT-treated rats (**F**). Hepatic protein expression of Col1a1 (**G**). Immunohistochemistry stainings of α-smooth muscle actin (α-SMA) and its quantification in all rat groups (**H**,**I**). Gene expression level of hepatic *α-SMA* in the rat groups (**J**). Representative Western blot of hepatic α-SMA in healthy, CCl_4_, control (CCl_4_ + WD), ST- and LT-treated rats (**K**). Gene expression levels of *Glial fibrillary acidic protein (GFAP)* in the livers of healthy, CCl_4_, control (CCl_4_ + WD), ST- and LT-treated rats (**L**). The scale bar is 200 µm in the 10× magnifications and 100 µm in the 20× magnifications. Results are expressed as means + SEM; n = 5 per group. *(*/** p* < 0.05/ *p* < 0.001 vs. Healthy; #/## *p* < 0.05/ *p* < 0.001 vs. CCl_4_; °/°° *p* < 0.05/*p* < 0.001 vs. control; + *p* < 0.05 vs. ST).

**Figure 3 cells-13-01707-f003:**
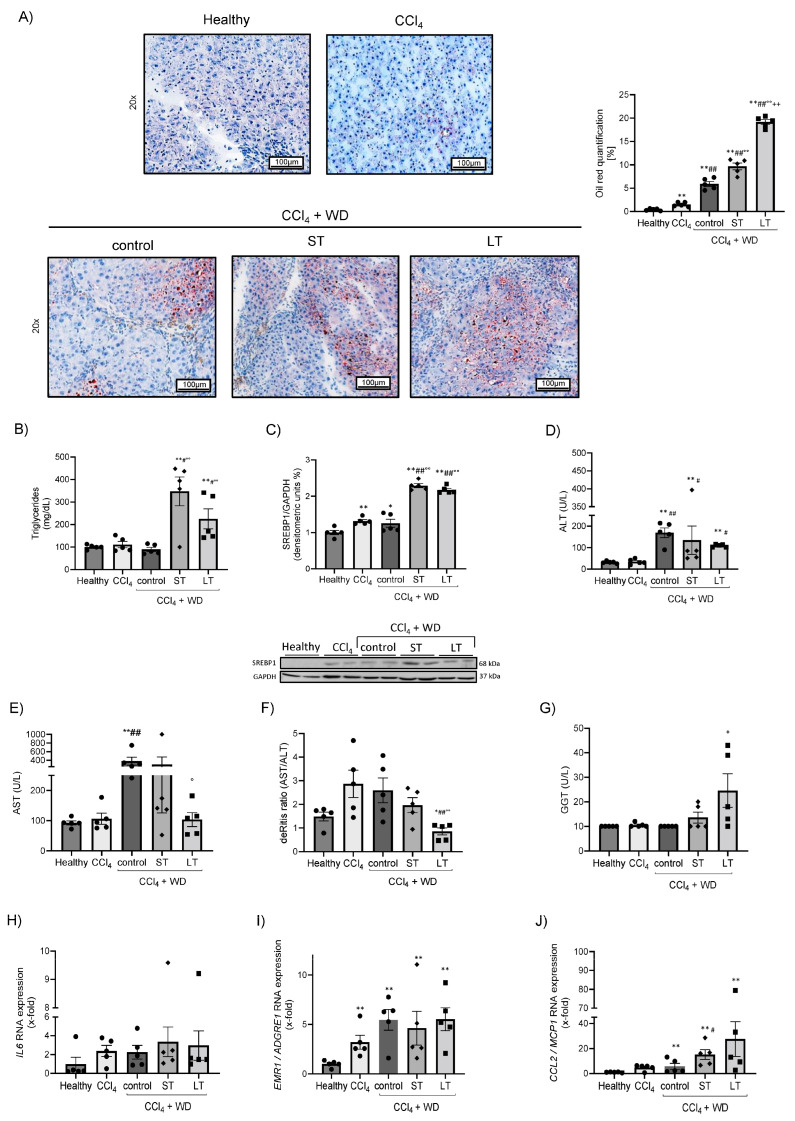
Hepatic steatosis and inflammation after ST and LT Phenobarbital treatment. Effect of carbon tetrachloride (CCl_4_), CCl_4_ + WD and ST or LT Phenobarbital treatment on liver steatosis in SD rats. Hepatic Oil red O staining of healthy, CCl_4_, control (CCl_4_ + WD), ST- and LT-treated rats and its quantification (**A**). Plasma triglyceride levels in all rat groups (**B**). Representative Western blot of the lipogenesis marker Sterol regulatory element-binding protein-1 (SREBP1) and its quantification (**C**). Plasma transaminase levels (ALT; AST) (**D**,**E**) and the deRitis ratio (**F**) in healthy, CCl_4_, control (CCl_4_ + WD), ST- and LT-treated rats. Gamma-Glutamyl Transferase (GGT) levels in all rat groups (**G**). Gene expression of pro-inflammatory markers *IL6*, *EMR1* and *CCL2* (**H**–**J**) in the livers of healthy, CCl_4_, control (CCl_4_ + WD), ST- and LT-treated rats. The scale bar is 100 µm. Results are expressed as means + SEM; n = 5 per group. (*/** *p* < 0.05/*p* < 0.005 vs. Healthy; #/## *p* < 0.05/*p* < 0.005 vs. CCl_4_; °/°° *p* < 0.05/*p* < 0.005 vs. control; ++ *p* < 0.005 vs. ST).

**Figure 4 cells-13-01707-f004:**
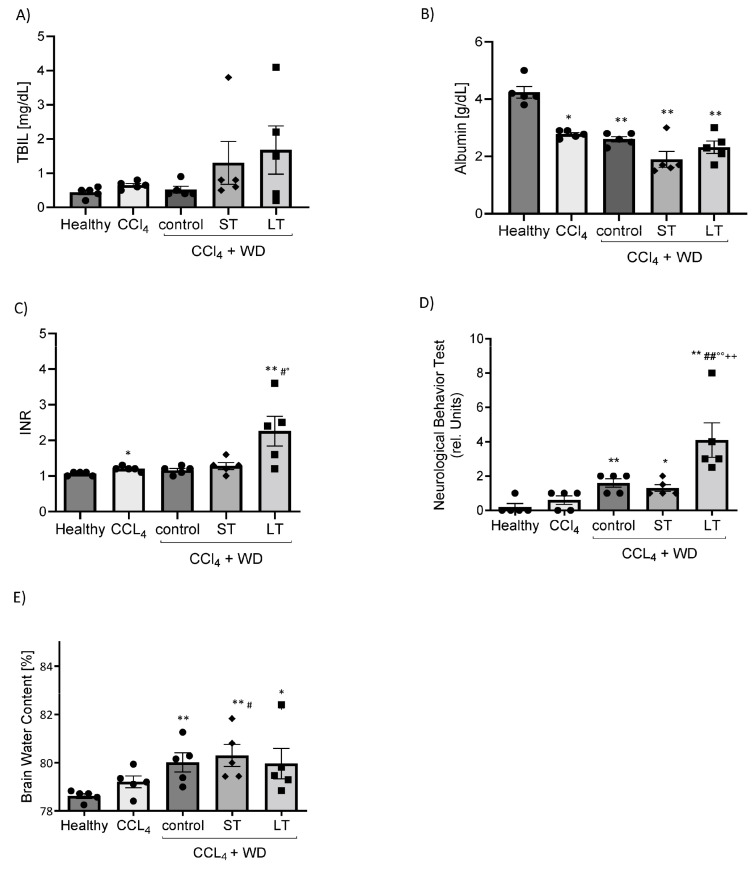
Phenobarbital-induced organ dysfunction of liver and brain in experimental cirrhosis. Effect of carbon tetrachloride (CCl_4_), CCl_4_ + WD and Phenobarbital on total Bilirubin (**A**), Albumin (**B**) and INR (**C**) in plasmas of SD rats. Extrahepatic impairment of the brain shown by the neurological behavior test (**D**) and the brain water content (**E**) in all rats. Results are expressed as means + SEM; n = 5 per group. (*/** *p* < 0.05/*p* < 0.005 vs. Healthy; #/## *p* < 0.05/*p* < 0.005 vs. CCl_4_; °/°° *p* < 0.05/*p* < 0.005 vs. control; ++ *p* < 0.005 vs. ST).

## Data Availability

Data are contained within the article and Appendix A.

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
