# Peer review of "Decompensated MASH-Cirrhosis Model by Acute and Toxic Effects of Phenobarbital"

_cells, 2024, doi:10.3390/cells13201707_

Round 1
Reviewer 1 Report
Comments and Suggestions for Authors
MASH is focused on as one of metabolic syndrome, and developing hepatitis is after steatosis. MASH-cirrhosis is developed after the developing steatohepatitis. The model animal of MASH-cirrhosis has not been developed, although the several animal models of MASH-steatohepatitis. This study indicates the novel animal model of MASH-cirrhosis in SD rats. This study is very useful of studying the MASH. However, several data of results were not suitable for publishing this manuscript. Authors should address below the concerns before publishing the manuscript.
Major
1. The histological data are too small to understand progression of steatohepatitis and cirrhosis. The photo data should be bigger size and indicate the low and high-magnification (Fig 2 and 3). Only these too small photo data cannot discuss the progression of steatohepatitis and cirrhosis.
2. All samples should be scored in according to hepatic fibrosis score. (e.g. https://link.springer.com/article/10.1007/s40290-019-00305-z)
3. The amount of triglyceride in the liver should be measured, because the photo data in control group was similar to one in LT group. The accurate quantification of TG makes the data more rigor.
4. Neurological behavioral test was performed to measure hepatic encephalopathy. But, phenobarbital may also affect the neurological behavior. If authors want to discuss the point, authors should compare the neurological behavior between ST or LT with CCl4+WD and ST or LT with normal diet.
5. In line 328-329, authors indicated that the vital liver mass was decreased in LT group compared with the others group. If authors want to discuss the point, the decrease of vital liver mass should be measured in time course.
Minor
1. In line 86-87, spaces are needed between numeral and unit (μg/g).
2. Fig. 1A is not enough informative. Endpoint (20 weeks?) in all groups and timing of administration of pentobarbital in ST group should be added.
Comments on the Quality of English LanguageThere are some errors in the manuscript. Please read above comment (Minor 1).
Author Response
Dear Reviewer,
please see our point by point answer in the attached document.
Thank you and best regards,
Sabine Klein

Reviewer 2 Report
Comments and Suggestions for Authors
Kraus et al used CCl4+WD+PB to induce MASH-Cirrhosis. They mentioned that this fast and reproducible rat model mimicking decompensated MASH-cirrhosis in humans.
1. In 2016, a similar diet and chemical-induced murine NASH, fibrosis and liver cancer model was reported in Journal of Hepatology by Tsuchida et al. This manuscript need to point out the unique or clinical useful viewpoint of their model as compared with Tsuchida et al, 2016. Why phenobarbital is necessary to use? Drink sugar water (Tsuchida et al, 2016) is more relevant to human lifestyle.
2. In Fig 1, supplement with phenobarbital short-term or long-term can cause mortality and moderate ascities was observed. These data indicated that this model may too toxically and severe. Indeed, the mortality and ascities are less in the early and middle stages of human MASH-cirrhosis.
3. In Fig 2B, the obvious Sirius red stain was observed in CCl4-treated mice, why a-SMA level were low in Fig 2F and G.
4. In Fig 3A, the obvious oil red o stain was observed in CCl4+WD-treated mice, why hepatic TG level was not increased in Fig 3B. SREBP1 is mainly regulate cholesterol level (Fig 3C). Another MASH-related character is inflammation, however, the related data still less in Fig 3H~J.
5. How about the glucose and Insulin-resistance status in this model?
Author Response
Dear Reviewer,
please see the attached document for our point by point answers.
Thank you very much and best regards,
Sabine Klein

Round 2
Reviewer 1 Report
Comments and Suggestions for Authors
Authors addressed to my comments, and revised manuscript is improved. However, a point should be improved, because reader could not understand histological changes between LT and another in Fig. 2A. I think that the histological changes including ballooning and inflammation in LT would be occur around Glisson's capsule. This manuscript did not indicate the high-magnification histological images in healthy, CCl4, control and ST groups to understand the histological changes in LT group. Authors should prepare the high-magnification histological images around Glisson's capsule in all groups to indicate the form of cells in referring to the below image.

Author Response
Response to Reviewer 1:
Major
- Authors addressed to my comments, and revised manuscript is improved. However, a point should be improved, because reader could not understand histological changes between LT and another in Fig. 2A. I think that the histological changes including ballooning and inflammation in LT would be occur around Glisson's capsule. This manuscript did not indicate the high-magnification histological images in healthy, CCl4, control and ST groups to understand the histological changes in LT group. Authors should prepare the high-magnification histological images around Glisson's capsule in all groups to indicate the form of cells in referring to the below image.
Dear Reviewer, thank you very much for this valuable comment. The histological changes after LT treatment with phenobarbital occur around the Glisson´s trias, that is correct and we regret that we missed to show this in Figure 2A.
In the re-scanned liver sections we chose images near the Glisson´s trias and labelled the portal vein (PV) and the bile duct (BD). Those images are now shown as Figure 2A and the corresponding legend has been changed as follows: “Hematoxylin and Eosin (H&E) stainings in livers of healthy, CCl4, control (CCl4 + WD), ST and LT treated rats (A). Illustrations of the glisson´s trias, to show hepatocyte ballooning (a) as well as steatosis and inflammation (b) in LT treated livers. Steatosis, inflammation and hepatocyte ballooning is indicated by arrows in 40 x magnification (PV = portal vein; BD = bile duct).”
We hope that Figure 2A is now more suitable for demonstrating the histological changes after LT treatments. And we thank you for this valuable comment, which improves our manuscript.
Reviewer 2 Report
Comments and Suggestions for Authors
Although this manuscript has some limitations, the overall data remain informative in the field.
Author Response
Reviewers comment:
Although this manuscript has some limitations, the overall data remain informative in the field.
We thank the reviewer very, very much and it is an honor for us to receive such positive feedback!!!
Best regards,
Sabine Klein
Round 3
Reviewer 1 Report
Comments and Suggestions for Authors
Authors addressed to my comments, but readers could not understand the changing of cell forms between healthy and another group. Before scanning the slide, background correction is needed. The background in all images of Fig. 2A is too blue. Furthermore, the resolution is too low to understand the historical changes. The area of images in Fig. 2A is smaller than previous manuscript. Author should correct the images for readers to understand the changes of cell forms in MASH models. Thus, I cannot recommend the publication for this journal until authors seriously address to tackle the above problems.
